# Conductors' *tempo* choices shed light over Beethoven's metronome

**Almudena Martin-Castro**[1]ⓞ, **Iñaki Ucar**[iD][2]ⓞ *

**1** Facultad de Ciencias, Universidad Nacional de Educación a Distancia, Madrid, Spain, **2** UC3M-Santander Big Data Institute, Universidad Carlos III de Madrid, Getafe, Spain

ⓞ These authors contributed equally to this work.
* inaki.ucar@uc3m.es

**Data Availability Statement:** All relevant data are within the manuscript and its Supporting information files.

**Funding:** The author(s) received no specific funding for this work.

## Abstract

During most part of Western classical music history, *tempo*, the speed of music, was not specified, for it was considered obvious from musical context. Only in 1815, Maelzel patented the metronome. Beethoven immediately embraced it, so much as to add *tempo* marks to his already published eight symphonies. However, these marks are still under dispute, as many musicians consider them too quick to be played and even unmusical, whereas others claim them as Bethoven's supposedly written will. In this work, we develop a methodology to extract and analyze the performed *tempi* from 36 complete symphonic recordings by different conductors. Our results show that conductor tempo choices reveal a systematic deviation from Beethoven's marks, which highlights the salience of "correct *tempo*" as a perceptive phenomenon shaped by cultural context. The hasty nature of these marks could be explained by the metronome's ambiguous scale reading point, which Beethoven probably misinterpreted.

## Introduction

The importance Beethoven gave to *tempo* as an essential component of his music idea is well documented. Indeed, he welcomed with enthusiasm the invention of the metronome by Johann N. Maelzel (Fig 1) and even attributed the success of his 9th Symphony to its newly added *tempo* marks [1]. The great paradox of this story is that, in spite of Beethoven's involvement, these marks have not helped clarify the *tempo* of his music. On the contrary, since their publication, they have long been debated and generally disregarded by performers [2–4]. Probably the most paradigmatic case is the Op. 106, also called *Hammerklavier* sonata, which starts with a decidedly unfeasible indication of 138 beats per minute for the half note. This and other incongruities have led many performers to ignore these figures and rely on other musical cues to determine the right *tempo*. But there are also those who, seeking historically accurate performances, claim Beethoven's marks as his supposedly written will. Today there is no album, essay, or concert criticism that fails to mention *tempo* choices when Beethoven is on the program.

**Competing interests:** The authors have declared that no competing interests exist.

Many scholars have argued on this matter from different points of view. In the 1980s, the Historically Informed performances (or HIP movement), defined by their intent to perform music in the manner of the musical era in which it was conceived, blamed Romanticism and Wagner's conducting school for slowing down Beethoven's music performances [3, 7]. Others have challenged the authenticity and subjective validity of the marks, arguing they do not convey Beethoven's intentions [3], or looking into their documentary sources for possible copy mistakes [8]. Temperley went even further to say that "Beethoven's marks are almost useless as guides to performance speeds", arguing that the *rubato* practice in the 19th century made it impossible to choose one *tempo* for a given piece [4].

The fact that not all marks share the same poor reputation has particularly puzzled musicologists. The most controversial and intriguing explanation is the one that focuses on the functioning of the metronome itself [9–11]. After all, Beethoven owned one of the first units of a newly invented device. Unfortunately, his own metronome was lost during an exhibition celebrated in Vienna in 1921 [10]. But there is documentary evidence that at least on two occasions the composer had to take it to the watchmaker due to its unsteady behaviour [1]. Would it not

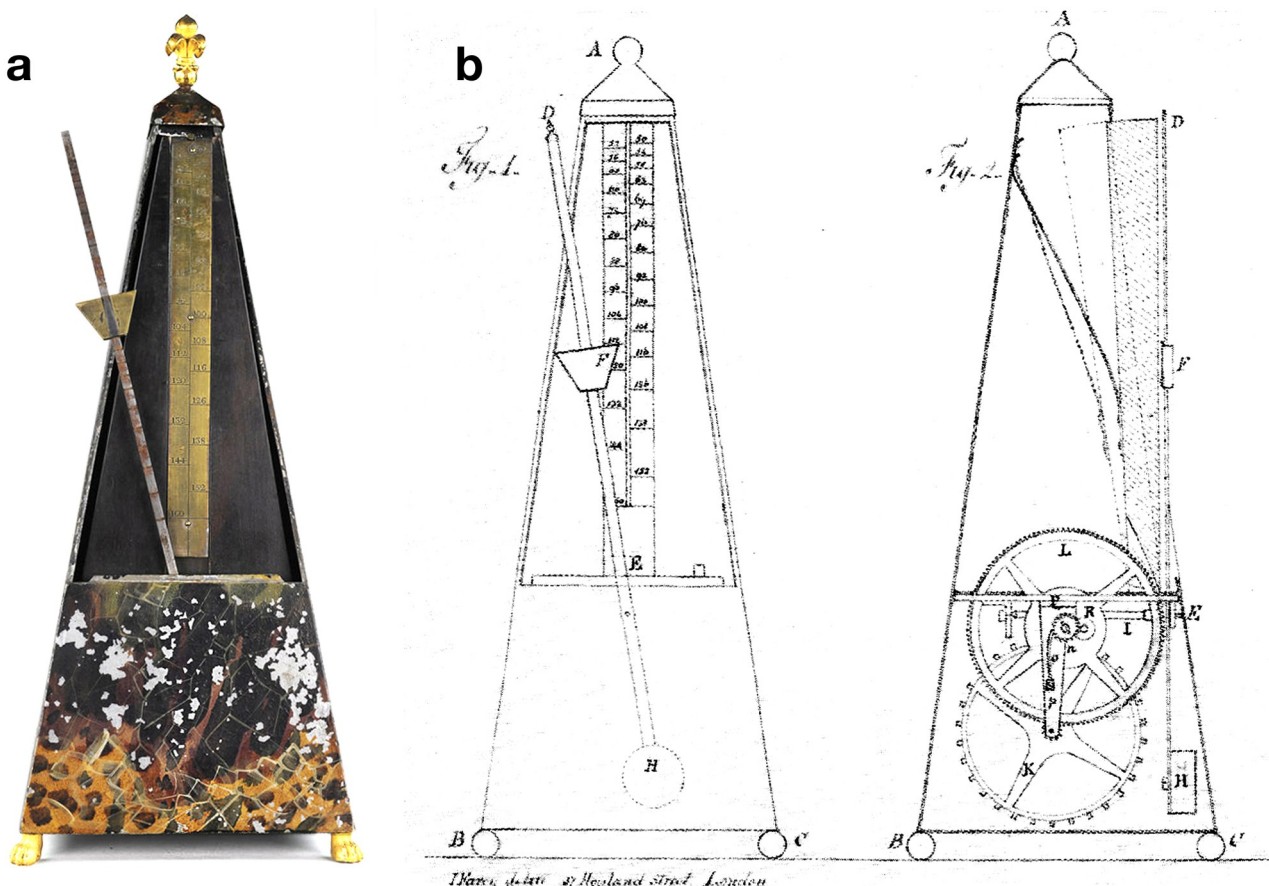

**Fig 1. Maelzel's metronome. a**, Metronome No. 7 from Tony Bingham's collection (TB 07) [5], made in Paris c.1816. **b**, Depiction from the 1815 English patent [6]. The metronome consists of two masses attached to a rod: the heaviest mass remains fixed at the lower end (hidden from view), while the upper mass (lighter, visible) can be moved along the rod to change the frequency of the oscillation. This way, the user can set up the desired *tempo* and determine its value by reading the scale behind the rod. The rod is fixed to the metronome's shaft and can oscillate around it. To compensate for friction, an impulse force is added to the system with the aid of a spring-driven escapement wheel, which also produces the characteristic audible ticks of the metronome. All this mechanism is held in a pyramid-shaped box that amplifies the metronome's sound and supports its scale. This is also the basic functioning of contemporary mechanical metronomes.

be possible that some mechanical damage slowed its mechanism at some point, forcing Beethoven to choose faster figures than the ones he really intended? Two previous studies have tried to analyze this hypothesis from a mechanical point of view [9, 11]. Nevertheless, both failed to compare their proposed models with the fact they intended to explain, that is: the disparity between Beethoven's marks and the performers chosen *tempi*.

On the basis of all this debate, the key question is whether music entails certain *tempo* that performers can estimate, or if it is instead an arbitrary choice that only the composer can reveal. Indeed, until the 19th century, composers did not have a way to quantify music speed objectively, and rather relied on qualitative indications (such as *Allegro*, *Andante*, *Grave*) and the performers' expertise to characterise their pieces. Even after the metronome was invented, composers such as Brahms or Mendelssohn disregarded its use as deemed useless, arguing that any musician should be able to infer the "correct *tempo*" for any piece [12–14]. In this regard, there is scientific evidence which suggests that *tempo* information is coded not only in melody representation and rhythm [15, 16], but also in other music attributes such as pitch, timbre [17] and event density [18]. As a result, the distinct combination of melody, harmony, rhythm, orchestration and notation of a particular piece may influence the perception of an optimal *tempo* between reasonable limits [19, 20].

By analyzing a set of different performances, our approach is ultimately based on the "wisdom of crowds" phenomenon [21]. If *tempo* is an arbitrary choice, we would not expect any clear pattern among Romantics, and HI performances should follow Beethoven's marks closely. On the other hand, if *tempo* is a perceptive phenomenon shaped by cultural context, Romantic performances would expose the underlying "correct" or perceptual *tempo*, whereas HI's deliberate effort to match Beethoven's indications would skew their choices. Moreover, if the metronome is to blame for this controversy, a large collection of conductors' *tempo* choices would reveal, on average, a systematic deviation from the original marks, which could be explained by analyzing the mechanics of Maelzel's metronome.

## Results

### A systematic analysis of Beethoven's symphonies performances

In this work, we analyzed the complete recordings of Beethoven's symphonies as performed by 36 different conductors from different styles and time periods, ranging from the 1940s to the 2010s. The symphonies are, without any doubt, Beethoven's most characteristic work and precious legacy. That is probably why, as soon as he was in possession of a metronome, he added *tempo* figures in great detail to all of them. Also, most renowned conductors choose to record them as a set, giving rise to a very complete and diverse collection of complete symphonic recordings. We have classified them as Historically Informed (HI), under HI influence and Romantics, based on the analysis of L. D. Young [7] and the performance reviews included with the recordings.

We measured the performed *tempi* of these recordings in an automated manner. To characterize its fluctuations, the audio files were sampled continuously using an overlapping sliding window and a *tempo* estimation algorithm [22, 23]. Classical music poses a major challenge for this kind of algorithm [24] due to its lack of a percussive base and its rhythmic complexity in general. Thus, we developed a methodology to rectify the algorithm output (Fig 2a–2c). Finally, the resulting data set was validated against a second set of manually-curated samples. After this process, we obtained a very accurate description of the performed *tempi* of Beethoven's symphonies, which supports previous qualitative analyses (Fig 2d).

Then, we analyze the distributions of performed *tempi* by metronomic mark: for each group of conductors, all the marks are reduced by the same amount on average (Fig 3a). Therefore, based on the median *tempi* for each mark, we fitted a multilevel linear model, with

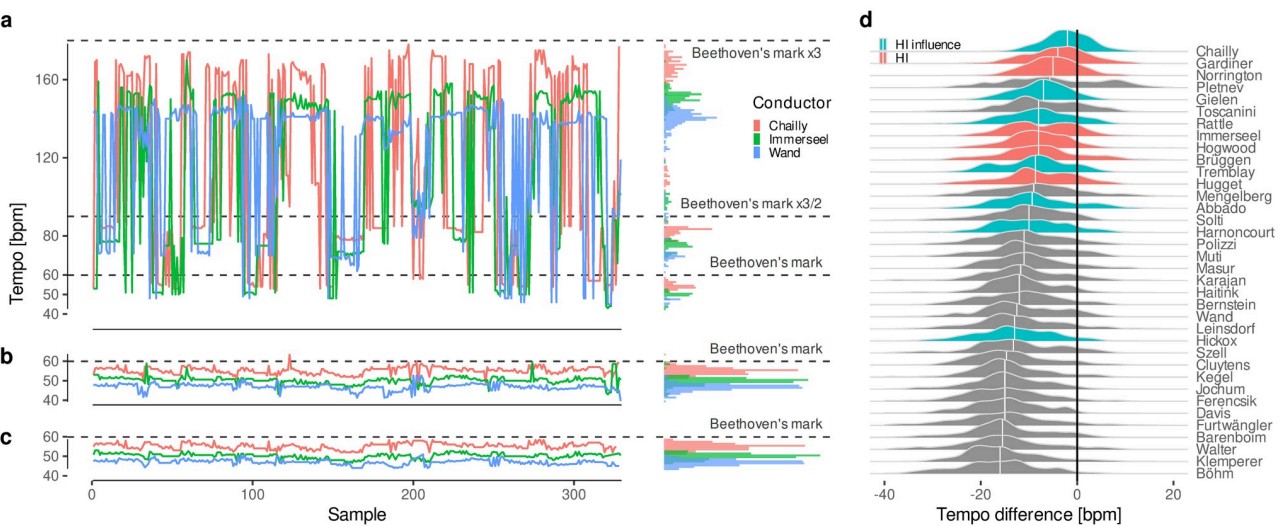

**Fig 2. *Tempo* data from symphonic recordings. a**, Representative example of raw data from the *tempo* extraction algorithm for 3 different conductors performing the 1st movement of the 3rd Symphony. Although the time series seem noisy on first sight, the histogram in the right panel shows a clear pattern: the algorithm not only detects the true *tempo* (components right below Beethoven's mark), but also multiples (or harmonics) of this frequency (in this example, x3/2 and x3). **b**, Using Beethoven's mark as a reference, harmonics in the raw data are found and rectified. **c**, A final smoothing ensures consistency in terms of continuity throughout contiguous samples. **d**, Distribution of *tempo* difference between conductors' *tempo* choices and Beethoven's marks. K. Böhm, at the bottom of the list, is well known among critics as one of the slowest performers of Beethoven [25]. On the other end, R. Chailly is the conductor who comes closer to the composer's indications as he reportedly intended. But even he falls slightly behind Beethoven's marks on average, a circumstance that has been even praised by some critics [26]. Remarkably, M. Pletnev has the most extreme and sparse distribution, reaching *tempi* far below and above other conductors. In fact, critics consider him an artist of contrasts, unorthodox and unpredictable [27].

the intercept as a random effect for each group of conductors (Fig 3b). The results show that HI, HI-influenced and Romantic conductors have slowed down Beethoven's marks by 6(2), 8 (2) and 13(2) bpm, respectively, on average. In the following, we consider the average discrepancy measured for Romantic conductors as the perceptual *tempo*.

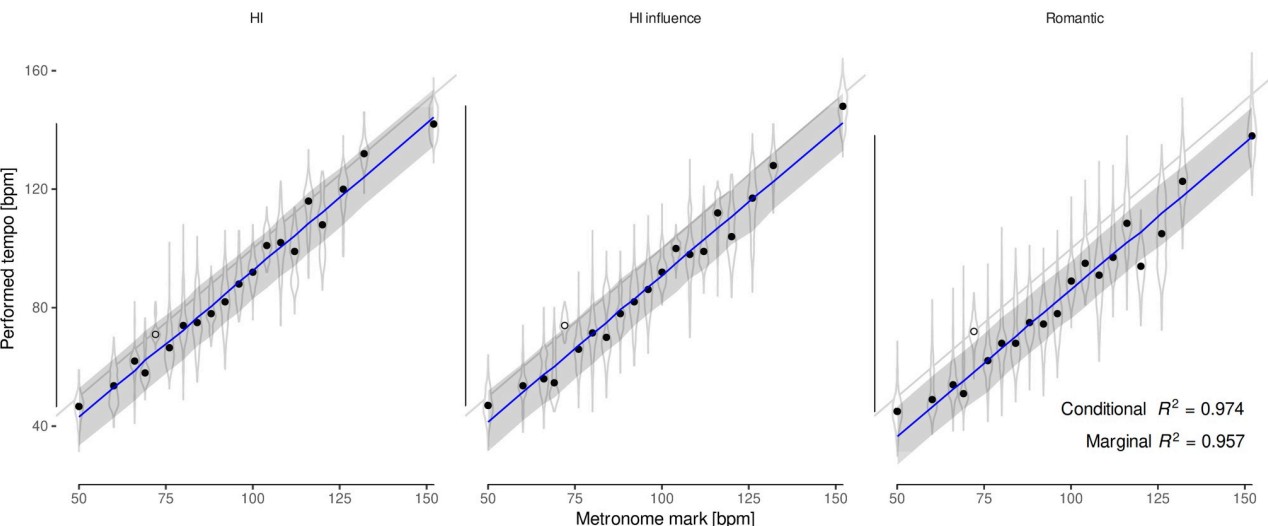

**Fig 3. Performed *tempo* by stylistic criterion vs. Beethoven's marks.** Each panel shows the distribution of *tempo* choices for each mark. The median for each distribution is shown as a dot, and the grayed line represents the 1:1 relation. On top of that, a mixed-effects regression line (in blue) for the medians, with a 95% Confidence Interval (CI), quantifies the effect of each group of conductors: all the marks are reduced on average by a fixed amount along the whole metronome range, preserving the relative discrepancy between groups. Interestingly, 72 bpm (7th Symphony, 4th movement; represented by an empty dot) seems to be the only mark that all groups accept as accurate, and therefore it was excluded from the regression model.

## Recreating Beethoven's metronome from photographs

We developed a mathematical model for the metronome based on a double pendulum, perfected with three kinds of corrections to take into account the amplitude of the metronome's oscillation, the friction of its mechanism, the impulse force, and, most importantly, the mass of its rod, which was neglected in previous studies. With the aid of this model, we then developed a methodology to estimate the original parameters of Beethoven's metronome from available photographs and the patent scheme [5]. A modern metronome was disassembled, measured and used to validate both the mathematical model and this methodology. Finally, we use our characterization of Beethoven's metronome to evaluate possible distortions, including the alteration of the lower mass [11] and friction [9] among others. We show that the only perturbation that causes the metronome to run homogeneously more slowly by 13(2) bpm, as Romantic performances suggest, is a displacement of the scale relative to the shaft of 16(3) mm.

## Discussion

Fig 2d shows that performed *tempi* are always slower than Beethoven's indications on average. The influence of the HI movement is also evident: attending to the median of their distribution, 12 out of the 15 fastest interpretations correspond to HI or HI-influenced performers, but they still fail to match Beethoven's marks. As Fig 3 shows, there is a systematic deviation from these marks that is homogeneous for the whole range of the metronome. Furthermore, the difference between groups is just a matter of degree, as though the HI movement had homogeneously increased the speed of all performances, preserving their relative discrepancies. This supports the hypothesis of an underlying perceptual *tempo*, as revealed by Romantic performances, that unwillingly affects HI performers as well, despite their conscious efforts to follow Beethoven's indications.

The homogeneous deviation of Romantic *tempo* choices from the marks can be explained by our metronome model, considering a displacement of the scale relative to the shaft of 16(3) mm. This could happen if, for some reason, its mechanism had fallen down within the box (when it was taken to repair, for instance) or if the scale was misplaced during its assembly. However, according to the patent scheme, there is little room in the box below the metronome, and the misplacement of the scale upwards would mean that the metronome had been poorly calibrated during its very construction, which wouldn't explain the disparity of the marks. There is another simpler explanation, though. By convention, the moving weight of the metronome must be placed below the mark it is meant to produce. Unfortunately, in the first metronomes, this weight was 15 mm high and had a triangular shape pointing downwards (Fig 4a). This could have led its users to read the metronome mark below the moving weight, instead of above. By jotting down the figures under this apparent arrow, Beethoven's marks would have resulted faster than he actually intended by, precisely, 12 bpm. Indeed, this is no accidental number: as we have shown, it is approximately the average difference between Romantic conductors' *tempo* choices and Beethoven's marks.

But could really Beethoven have committed such a mistake? In the first page of his autograph of the 9th Symphony, there is a revealing inscription from his own hand: "108 or 120 Maelzel" (Fig 4b). Some scholars have interpreted this text as proof of Beethoven's poor state of mind, his indecisiveness or some preliminary *tempi* range still to be decided [10, 29, 30]. But the big difference between these two figures make such hesitation unlikely for a composer who so often insisted on the importance of *tempo* as an essential part of his music. Moreover, if Beethoven had wanted to delimit a possible *tempi* range, he would have written "108-120", not "or" [10]. As we have clarified in this work, the distance between 108 and 120 on the scale, 15

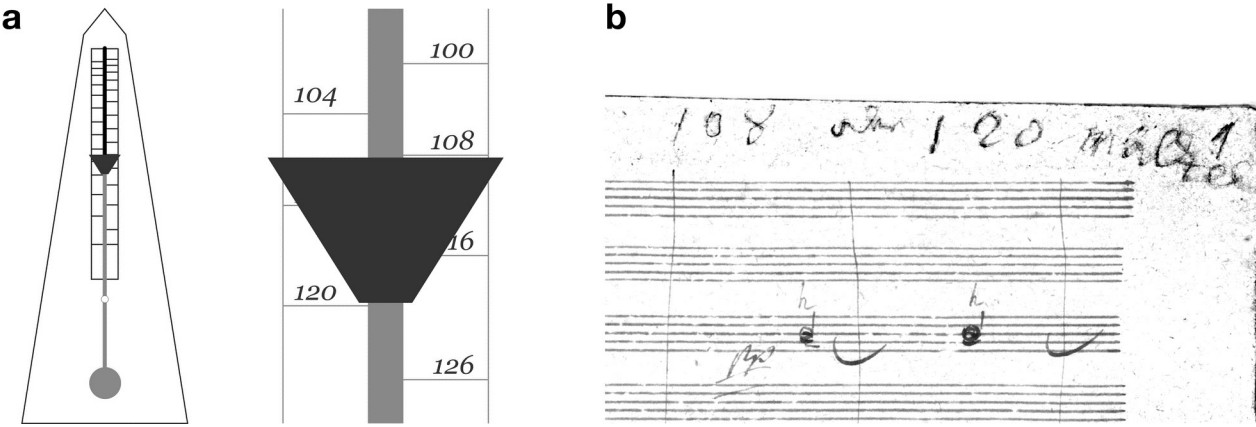

**Fig 4. Metronome's ambiguous reading point. a**, Diagram of the metronome and detail of the moving weight. This weight was 15 mm high, a distance equal to 12 bpm on the *tempo* scale throughout all its range. 44 out of 63 marks used by Beethoven could have been mistaken by another Maelzel mark exactly 12 bpm quicker. **b**, Enhanced image of Beethoven's inscription on the first page of the 9th Symphony autograph [28]: "108 oder 120 Mälzel", where "oder" means "or" in German, and "Mälzel" refers to Maelzel's metronome.

mm, matches exactly the size of the metronome's moving weight. This innocent annotation constitutes written proof that, after years using the metronome, there was a moment, at least, when Beethoven was not sure about how to read it. He even left his doubts annotated on the score, instead of using other methods to dispel them.

This could also explain why not all of his marks are usually dismissed. Perhaps Beethoven was confused at times, for his lack of experience using the device. Or maybe, the differences originated in the user, the person actually holding the device. We know from the conversation books that people used to communicate with Beethoven in his later years, that it was his nephew, Karl, who jotted down the *tempi* of the 9th Symphony while the composer rehearsed it on the piano [8]. Beethoven could have required some help to measure his first symphonies as well, leading to different readings of the metronome. Indeed, Anton Schindler, Beethoven's first biographer and secretary, was also the first to discredit the composer's marks, insisting that he had needed to review some of them, bewildered by their apparent inconsistencies over time [31]. Schindler has been criticised later for his proven forgeries and general malpractice [32], but maybe there was some truth in these assertions.

In summary, our work, based on the analysis of 36 complete recordings of symphonic works, highlights the salience of the perceptual *tempo* as a product of idiomatic cues within music, as psychological research suggests. In a new illustration of the social phenomenon known as "wisdom of crowds", we have found that performers' median *tempo* choices follow a systematic deviation from Beethoven's marks. Furthermore, our accurate mathematical model of Maelzel's metronome, rules out the hypothesis of Beethoven's broken metronome and sheds light over a 200-year-old controversy among critics, performers and scholars. The most probable hypothesis is that Beethoven or his assistant misread the device, which should not be taken as a foolish mistake, but as a symptom of a design that had yet to be perfected, and that still lacked the cultural context to support its new users.

Above all, our work provides a methodology for data-based systematic analyses of contemporary recordings and classical music performances. This will allow musicologists and other scholars to have a new quantitative insight into a research field which usually relies on qualitative analyses mostly. Moreover, our findings regarding Beethoven's works in particular provide very valuable information for musicians and performers which will be able to look at the composer's *tempo* choices from a new perspective, reanalyze their individual validity and

apply the emergent criteria not only to Beethoven's symphonies, but also to all of his other metronomized works.

## Methods

### Data set

In this work, we selected 36 recordings of Beethoven's complete symphonic works as performed by 36 different conductors, and classified them as Historically Informed (HI), under HI influence and Romantic (Table 1). By convention, HI performances are those that use period instruments and follow all the usual HI stylistic criteria, whereas those considered HI-influenced may be not so strict in terms of instrumentation. Finally, Romantic performances are those previous to the 1980s, or more generally, those that do not adhere to HI performing criteria. The information necessary to complete this classification was gathered from the performance reviews included with the recordings and the analysis made by L. D. Young [7].

The 9th Symphony is exceptional for various reasons. First, it was metronomized seven years later than the others using a different device, of which the date of purchase is not known [10]. Second, its complexity, especially regarding the fourth movement, makes *tempo* extraction too unstable and unreliable. Finally, some authors have questioned the validity of the documentary sources where these *tempi* were first published, only months before the composer's death, due to multiple copy mistakes [8]. For all these reasons, we decided to exclude it from the analysis. The rest of the data set comprises 1188 audio files (one symphonic movement per track), more than 169 hours of music.

### *Tempo* extraction

Audio files were sampled using a sliding window. Its duration was defined as a fraction of the track, so that the average width was 30 seconds, with a 90% overlap. In this way, each symphonic movement is divided in the same number of samples, regardless of the interpreter and the duration of the track. Every sample was then analyzed using a state-of-the-art *tempo* extraction algorithm [24] that bases pulse detection on self-similarity relations within the rhythm of a musical recording [22, 23], and is implemented as part of the open-source framework Marsyas [33]. Sections containing a change of *tempo* or meter were identified and located on the score and the resulting samples. Different sections and movements were classified according to their meter (duple or triple meter, simple or compound). This classification is important in order to identify the most probable *tempo* harmonics detected by the *tempo* extraction algorithm for each sample (Figs 5 and 2a).

Then, data are grouped by conductor, symphony, movement and section. We compute a histogram for each group and locate its peak, which corresponds to the most detected *tempo* in each recording. These peaks are compared with Beethoven's metronome mark and its harmonics, taking into account the music meter. If the peak matches any of the harmonics, it is corrected accordingly (its value is divided by the corresponding harmonic). Corrected peaks are then used as a reference to correct all the *tempo* values in the recorded piece. The process is similar to the previous step: if a *tempo* sample matches one of the peak harmonics within a certain tolerance, it is divided by the value of the harmonic (Fig 2b). Tolerances are defined case per case to avoid harmonics overlap. Then, *tempo* values are corrected using a continuity criterion. In a typical recording, *tempo* can vary a lot, so the harmonics correction based on the histogram peak might sometimes fail. In those cases we can take advantage of the fact that *tempo* usually varies smoothly: each data point is compared with the previous 3 samples in search for the same harmonic relationships as in the previous step, and corrected appropriately if found.

**Table 1. List of recordings studied in this work.**

| Conductor | Orchestra | Recording | Label | UPC | Style |
|---|---|---|---|---|---|
| Abbado, Claudio | Berliner Philharmoniker | 2000-2001 | DG | 028947758648 | HI influence |
| Barenboim, Daniel | West-Eastern Divan Orchestra | 2011 | Decca | 028947835110 | |
| Bernstein, Leonard | Wiener Philharmoniker | 1977-1979 | DG | 028947492429 | |
| Böhm, Karl | Wiener Philharmoniker | 1969-1972 | DG | 028947919490 | |
| Brüggen, Frans | Orchestra of the 18th Century | 1984-1992 | Decca | 028947874362 | HI |
| Chailly, Riccardo | Gewandhausorchester Leipzig | 2007-2009 | Decca | 028947834922 | HI influence |
| Cluytens, André | Berliner Philharmoniker | 1957-1960 | Erato | 5099964830353 | |
| Davis, Colin | Staatskapelle Dresden | 1995 | Philips | 028947568834 | |
| Ferencsik, Janos | Hungarian State Orchestra | 1969-1976 | Hungaroton | 5991810401321 | |
| Furtwängler, Wilhelm | Philharmonia Orchestra Berliner Philarmoniker Wiener Philarmoniker Philharmonisches nnStaatsorchester Hamburg | 1947-1954 | Andromeda | 3830257490937 | |
| Gardiner, John Eliot | Orchestre Révolutionnaire nnet Romantique | 1991-1994 | DG | 028943990028 | HI |
| Gielen, Michael | SWR Sinfonieorchester nnBaden-Baden Freiburg | 1997-2000 | Hänssler | 4010276025078 | HI influence |
| Haitink, Bernard | Royal Concertgebouw Orchestra | 1985-1987 | Philips | 0028944207323 | |
| Harnoncourt, Nikolaus | Chamber Orchestra of Europe | 1990-1991 | Teldec | 0809274976826 | HI influence |
| Hickox, Richard | Northern Sinfonia of England | 1984-1988 | Resonance | 0680125050427 | HI influence |
| Hogwood, Christopher | The Academy of Ancient Music | 1985-1989 | Decca | 028945255125 | HI |
| Hugget, Monica & nnGoodman, Roy | The Hanover Band | 1982-1988 | Nimbus | 0710357514425 | HI |
| Immerseel, Jos Van | Anima Eterna Orchestra | 2005-2007 | Zigzag | 3700551732197 | HI |
| Jochum, Eugen | Concertgebouw Orchestra | 1967-1969 | Philips | 0028947581475 | |
| Karajan, Herbert von | Philharmonia Orchestra | 1951-1955 | Warner | 5099951586324 | |
| Kegel, Herbert | Dresdner Philharmonie | 1982-1984 | Capriccio | 4006408500001 | |
| Klemperer, Otto | Philarmonia Orchestra | 1960 | Arts | 0017685125225 | |
| Leinsdorf, Erich | Boston Symphony Orchestra | 1961-1969 | RCA | 0886919168228 | |
| Masur, Kurt | Leipzig Gewandhausorchester | 1972-1975 | Philips | 0028947527220 | |
| Mengelberg, Willem | Royal Concertgebouw Orchestra | 1940 | Archipel | 4035122401929 | |
| Muti, Riccardo | Philadelphia Orchestra | 1985-1988 | Warner | 5099909794627 | |
| Norrington, Roger | London Classical Players | 1987-1990 | Erato | 5099908342324 | HI |
| Pletnev, Mikhail | Russian National Orchestra | 2007 | DG | 0028947764090 | |
| Polizzi, Antonino | Prague Symphony Orchestra Budapest Symphony Orchestra | 1986-1994 | Polymnie | 3576079901205 | |
| Rattle, Simon | Wiener Philharmoniker | 2002 | EMI | 5099991562425 | HI influence |
| Solti, Georg | Chicago Symphony Orchestra | 1986-1989 | Decca | 0028943040020 | |
| Szell, George | Cleveland Orchestra Chorus | 1956-1964 | Sony | 0888837371520 | |
| Toscanini, Arturo | NBC Symphony Orchestra | 1949-1952 | RCA | 0828765570220 | |
| Tremblay, Jean-Philippe | Orchestre de la Francophonie | 2009 | Analekta | 0774204997526 | HI influence |
| Walter, Bruno | Columbia Symphony Orchestra | 1958-1959 | Sony | 5099750231227 | |
| Wand, Günter | NDR Symphony Orchestra | 1985-1988 | RCA | 0743218910920 | |

Each recording details the conductor's name, orchestra, recording dates, label, Unique Product Code (UPC) and style.

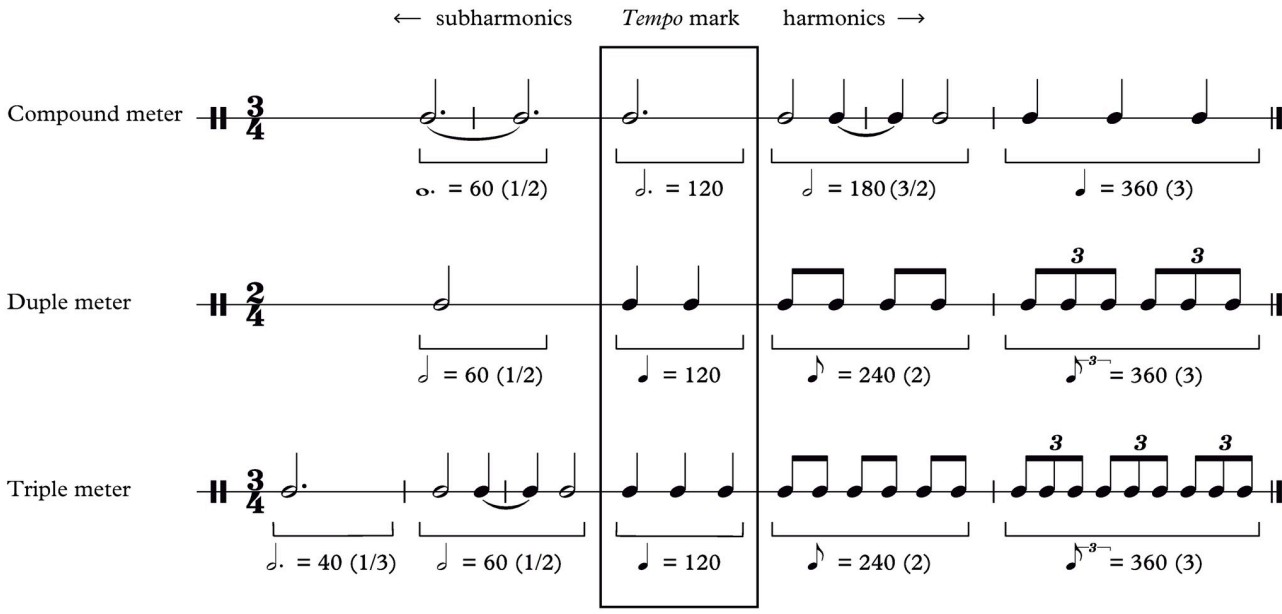

**Fig 5. Most common *tempo* harmonics for each kind of meter.** The *tempo* extraction algorithm relies on periodic patterns and rhythmic self-similarities. This explains why many of its estimated *tempi* are actually multiples or submultiples of the real *tempo* of the sample. In this work, we have called these kinds of mistaken *tempi* "harmonics" due to the similarity with the homonym physical phenomenon. Their most common values depend on the metric structure of the music and are displayed here. More rarely, we also detected: (i) harmonics 2 y 3/4 in compound meters; (ii) harmonics 2 y 3/4 in simple meters due to the occasional use of triplets; (iii) harmonic 2/3, in simple triple meters.

Finally, outliers, defined as data points that differ more than 2 standard deviations from the corrected peak, are removed and replaced by interpolated values (Fig 2c).

A complementary methodology was developed to assess the validity of this collection of *tempo* measurements. We sampled 30 seconds from the last minute of every movement, thus compiling a set of *finales*, where *tempo* is arguably more stable. *Tempo* was also extracted using Marsyas on first pass, but then carefully curated by hand. The main data set is validated by comparing the median *tempo* for each conductor and mark with the median *tempo* as obtained from this data set of finales (Fig 6).

## Metronome model

Contemporary mechanical metronomes preserve essentially the same design as Maelzel's metronome (Fig 7). The angular frequency of oscillation, Ω, is obtained as a function of three multiplicative terms:

$$\Omega = f_{\text{ang}}^{-1}(\theta) \cdot f_{\text{fric}}^{-1}(\epsilon) \cdot \sqrt{g \frac{M'R - \frac{\mu'}{2}(l - L) - r}{M'R^2 + \frac{\mu'}{3}(L^2 + l^2 - lL) + r^2}} \qquad (1)$$

where the last term draws from the classical expression for an ideal double pendulum, but includes corrections to account for the non-negligible mass, *μ*, of the rod. Other parameters are the gravitational acceleration (*g*), the nondimensionalized lower ($M' = M/m$) and rod ($\mu' = \mu/m$) masses, the distances of the lower and upper masses to the shaft (*R* and *r*, respectively), and the length of the two ends of the rod from the shaft (*L* and *l*, respectively). The first two

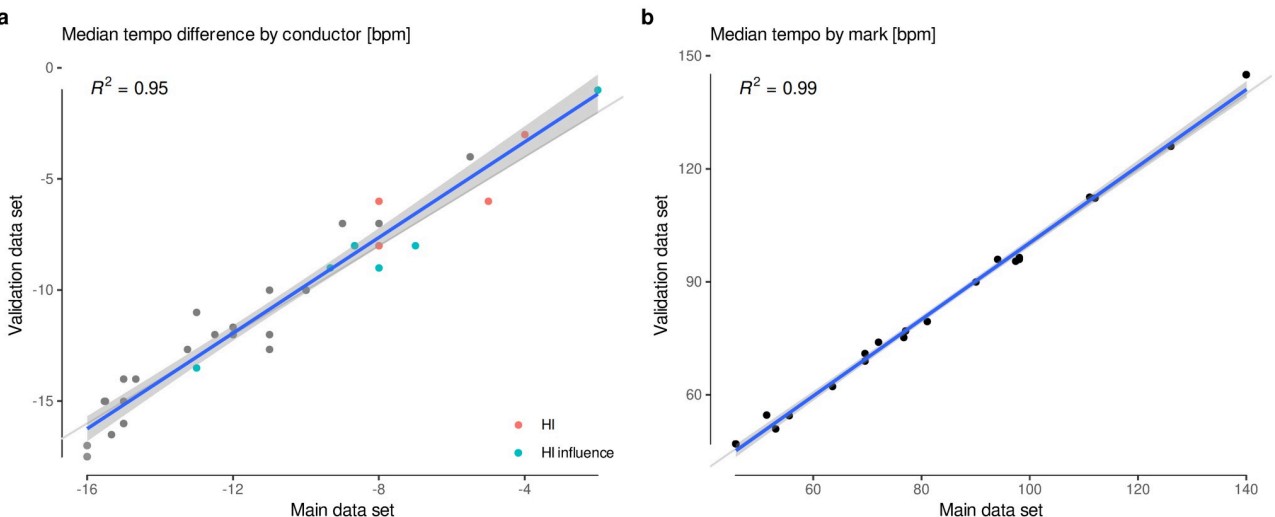

**Fig 6. Validation of *tempo* data. a**, Each dot represents a conductor, and compares the median *tempo* difference (*tempo* choice minus Beethoven's mark) for the main and validation data sets. **b**, Each dot represents a metronome mark, and compares the median *tempo* for the main and validation data sets. Both figures show a 1:1 relation, which ensures the consistency of the main data set.

terms, $f_{ang}$ and $f_{fric}$, are further corrections to account, respectively, for large oscillations (usually ranging from $\theta = 40°$ to $60°$) and friction and impulse forces:

$$f_{ang}(\theta) = 1 + \sum_{n=1}^{\infty}\left[\frac{(2n-1)!!}{(2n)!!}\sin^{2n}\left(\frac{\theta}{2}\right)\right]^2 \tag{2}$$

$$f_{fric}(\epsilon) = 1 + \frac{1}{\pi}\sin^{-1}\left(\frac{\epsilon}{1-\epsilon}\right) - \frac{1}{\pi}\sin^{-1}\left(\frac{\epsilon}{1+\epsilon}\right) \tag{3}$$

where $\epsilon$ is a nondimensional parameter that must range from $0 \leq \epsilon \leq 0.5$, so that the equation has a real solution (i.e, the metronome oscillates) [9]. We determined that $\epsilon = \tau/(\Omega^2 I\theta)$ is proportional to the friction torque $\tau$, and inversely proportional to the angular frequency squared and the moment of inertia $I$.

A contemporary metronome (Fig 7a) was used to validate the model. First, the angular frequency for each metronome mark was measured by means of extracting the tickling period over 15-second audio samples. Then, the metronome was dismantled and all parameters were measured (dimensions and masses; Table 2). Our model achieves even better accuracy than the calibration set by the manufacturer (MAE of less than 2 bpm, compared to a MAE of 3 bpm for the metronome scale; Fig 8a).

The same contemporary metronome was used to study the effect of each kind of correction. To this end, the true mass of the rod, the true oscillation angle and the maximum friction allowed by the model ($\epsilon = 0.5$) were separately compared against the null model (null mass, oscillation angle and friction) along the whole scale range (Fig 8b). As expected, the mass of the rod contributes the most to the model accuracy, and the effect of friction is negligible except for the lowest oscillation frequencies.

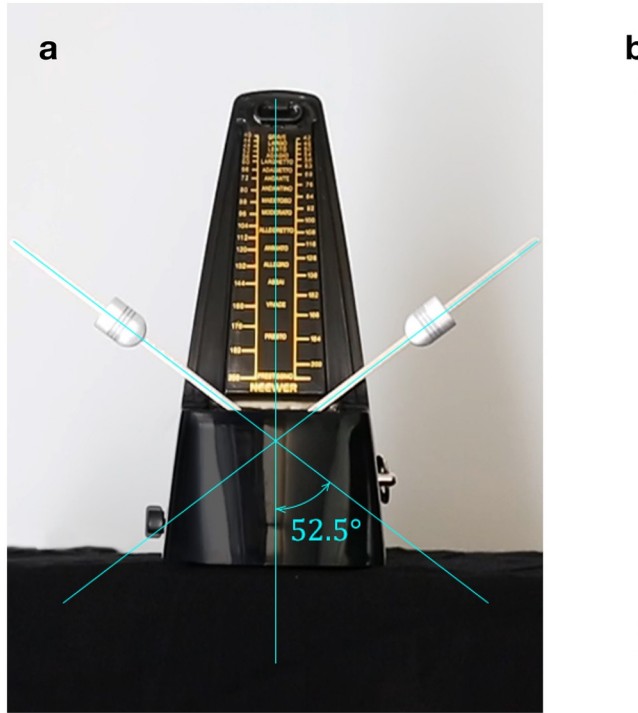
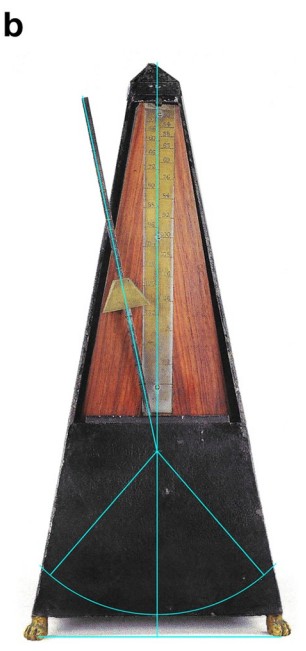
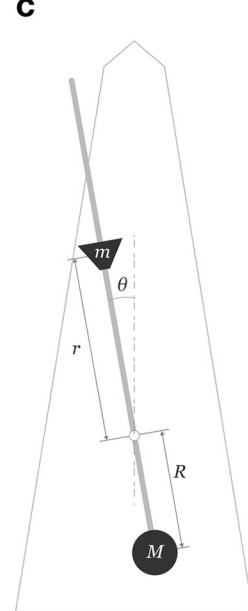

**Fig 7. Metronomes. a**, Contemporary metronome used as a control, model Neewer© NW-707. The maximum angle of oscillation was measured by recording the metronome's motion and creating this composite of two video frames. **b**, Metronome No. 6 from Tony Bingham's collection (TB 06) [5], sold in London, but almost certainly made in Paris c.1816. Auxiliary lines were added to the photographs to locate the shaft and the maximum oscillation angle. The lower mass is estimated to hang 2 cm above the bottom of the box, according to the patent scheme. **c**, Metronome diagram. The metronome is based on a double pendulum, where the heaviest mass, *M*, remains fixed at the lower end of a rod, and the lighter mass, *m*, can be moved upwards and downwards to change the oscillation frequency. The distances from the shaft to each center of mass are designated by *R* and *r*. *θ* is the pendulum's angle of oscillation.

## Model transformation and fit

Neglecting the effect of friction ($f_{\mathrm{fric}} \approx 1$), we express $\Omega^2$ as a linear combination of polynomial terms of *r*:

$$\Omega^2 = a_0 + b_2 \left( \frac{g}{f_{\mathrm{ang}}^2(\theta)} r + \Omega^2 r^2 \right) \tag{4}$$

**Table 2. Measurements for all the metronomes considered.**

| Metronome | Dimensions [mm] | | | | Angle | Masses [g] | | |
|---|---|---|---|---|---|---|---|---|
| | *h* | $r_{\mathrm{cm}}$ | *l* | *R* | *θ* [°] | *M* | *m* | *μ* |
| Control | 200(1) | -4(1) | 138.0(3) | 36.4(3) | 52.5(3) | 31.01(1) | 7.10(1) | 3.59(1) |
| Control (photo) | 200(1) | -8(1) | 137(1) | 36(1) | 52.5(3) | - | - | - |
| TB 06 | 310(1) | -4.6(4) | 195(3) | 68(1) | 40(5) | - | - | - |
| TB 07 | 332(1) | -5.0(5) | 198(2) | 65(1) | 40(5) | - | - | - |
| Patent | 310(1) | -5.0(5) | 190.3(6) | 63(1) | 40(5) | - | - | - |

A contemporary metronome (Fig 6a) was used as a control: first, with precise measurements from a dismantled unit, including the masses; second, from a photograph, following the same procedure used for the patent (Fig 1b) and the old metronomes TB 06 and TB 07 (Figs 6b and 1a). The total height *h* was used to calibrate the measurement process. The distance $r_{\mathrm{cm}}$ is the shift of the center of mass of the moving weight with respect to the top of the weight, which is needed to measure *r* for each metronome mark.

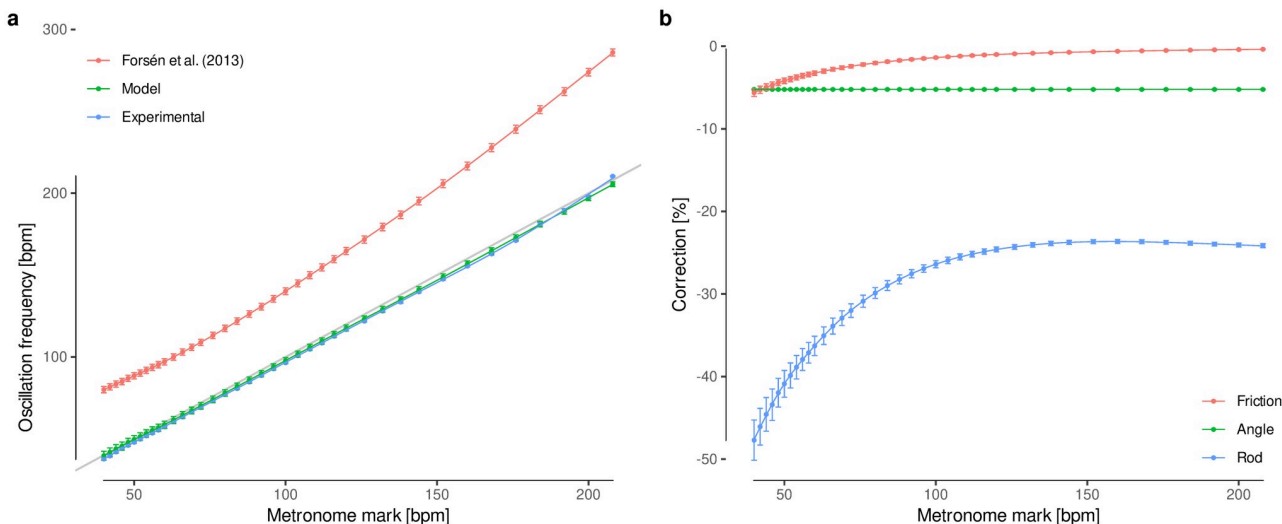

**Fig 8. Metronome model. a**, Model validation. The parametrization of a contemporary metronome is compared to its experimental oscillation frequency. It should be noted that the experimental results do not exactly follow the 1:1 relation (gray line), which means that the calibration of the scale has a small error, and our model accurately predicts it. The model by Forsén *et al.* (2013) [11], which uses a double pendulum without corrections, is included for completeness. **b**, Effect of corrections throughout the whole range for the same metronome, expressed as a percentage over the null model (frictionless, small-angle approximation for a massless rod) for each metronome mark.

where

$$a_0 = \frac{g}{f_{\mathrm{ang}}^2(\theta)} \cdot \frac{M'R - \frac{\mu'}{2}(l - L)}{M'R^2 + \frac{\mu'}{3}(L^2 + l^2 - lL)} \tag{5}$$

$$b_2 = -\frac{1}{M'R^2 + \frac{\mu'}{3}(L^2 + l^2 - lL)} \tag{6}$$

This linear model was fitted for two metronomes dated 1816 in Bingham's collection [5], similar to Beethoven's device (Figs 2a and 7b), the patent diagram (Fig 2b), and the contemporary metronome (Fig 7a) as a control (Fig 9a). Metronome dimensions were measured using Fiji [34, 35] on the basis of the total heights reported in Bingham's catalogue (Table 2). The total height is assumed to be 31 cm for the patent according to the patent description and the height of the oldest metronome (Fig 2a). The oscillation angle is taken as the maximum inclination, bounded by the box. Parameter *R* cannot be directly measured for some metronomes (when the box hides the lower mass), so it was estimated taking into account the box size and the patent description. Given that the lower mass hangs approximately from the end of the rod, it is assumed that $L \approx R$. With these assumptions, we estimated the nondimensional masses, $M'$ and $\mu'$, for each metronome from the regression coefficients (Fig 9b). Results show that this methodology accurately estimates the masses for the control metronome, and thus, in the following we take the averages of the old metronomes and the patent as a parametrization of Beethoven's metronome: $M' = 4.0(1)$ and $\mu' = 0.64(3)$, with the rest of the parameters equal to the measurements for the patent (Table 2).

## Performed *tempo* vs. metronome distortions

Performed *tempo* is modelled as a function of the metronome marks by means of a mixed-effects linear model, using the intercept as a random effect for each conducting style (Fig 3).

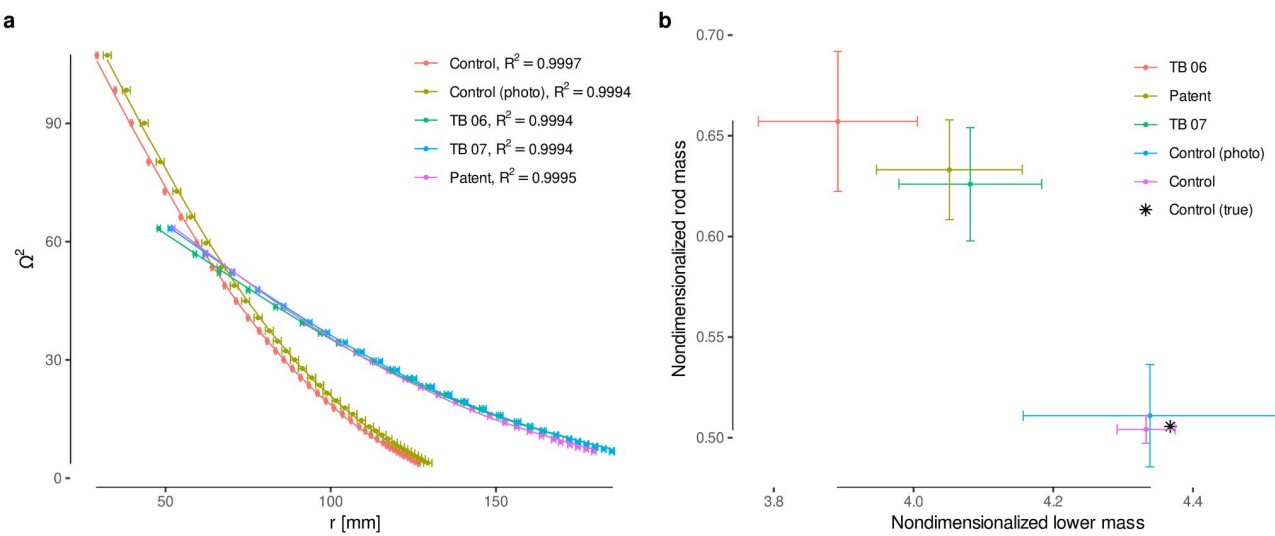

**Fig 9. Parameter estimation for all the metronomes considered. a**, Model fit for the oscillation frequency squared as a function of the position of the moving weight. **b**, Estimation of nondimensionalized masses $\mu'$ (rod) and $M'$ (lower mass). Both controls (measuring a dismantled metronome with precision as well as measuring all the distances from a photograph) accurately estimate the true masses for the contemporary metronome, thus validating the estimation for the rest of the metronomes.

This model reveals a common trend shared by all groups: a shifted 1:1 relationship with the marks (Confidence Interval: 95% CI [0.95, 1.03]), and a significative random effect (Likelihood Ratio Test: LRT = 15.29, $p$ <.001), which suggests that performers slow down Beethoven's marks, on average, by a fixed amount that is different for each group of conductors. Hereafter, we consider the average discrepancy measured by this model for Romantic conductors as a proxy for Beethoven's intended *tempo*. Thus, we are interested in comparing these results with possible distortions that decrease the metronome's frequency by a comparable amount throughout all its range, without remarkable defects or anomalous behaviors that could have warned Beethoven about a flaw in the device.

We analyzed the variation of the lower mass $M$ and its distance to the shaft $R$ resulting from some possible blow that could have broken or loosen it up, as proposed by Forsén *et al.* (Fig 10a and 10b). However, these are similar distortions that mostly affect the slower frequencies. We also considered different inclinations of the metronome, maybe held in an unstable position on the piano while rehearsing. This decreases the gravitational acceleration experimented by the pendulum, but would have caused the quicker frequencies to decelerate mostly and, more importantly, would only be noticeable for extremely sharp inclinations (Fig 10c). We also analyzed an increase of friction resulting from poor lubrication, but as shown previously, its effect is negligible for higher frequencies (Fig 8b) and, when increased, causes the metronome to stop completely at lower frequencies [9]. Finally, a shift of the moving weight relative to the scale is the only mechanism that describes the observed slow-down of *tempo* by performers, which in turn can be explained by the user reading the marks below the moving weight (Fig 10d).

## Supporting information

**S1 File. Supporting data and methods.** The `bmetr` R package contains all supporting data and methods.
(GZ)

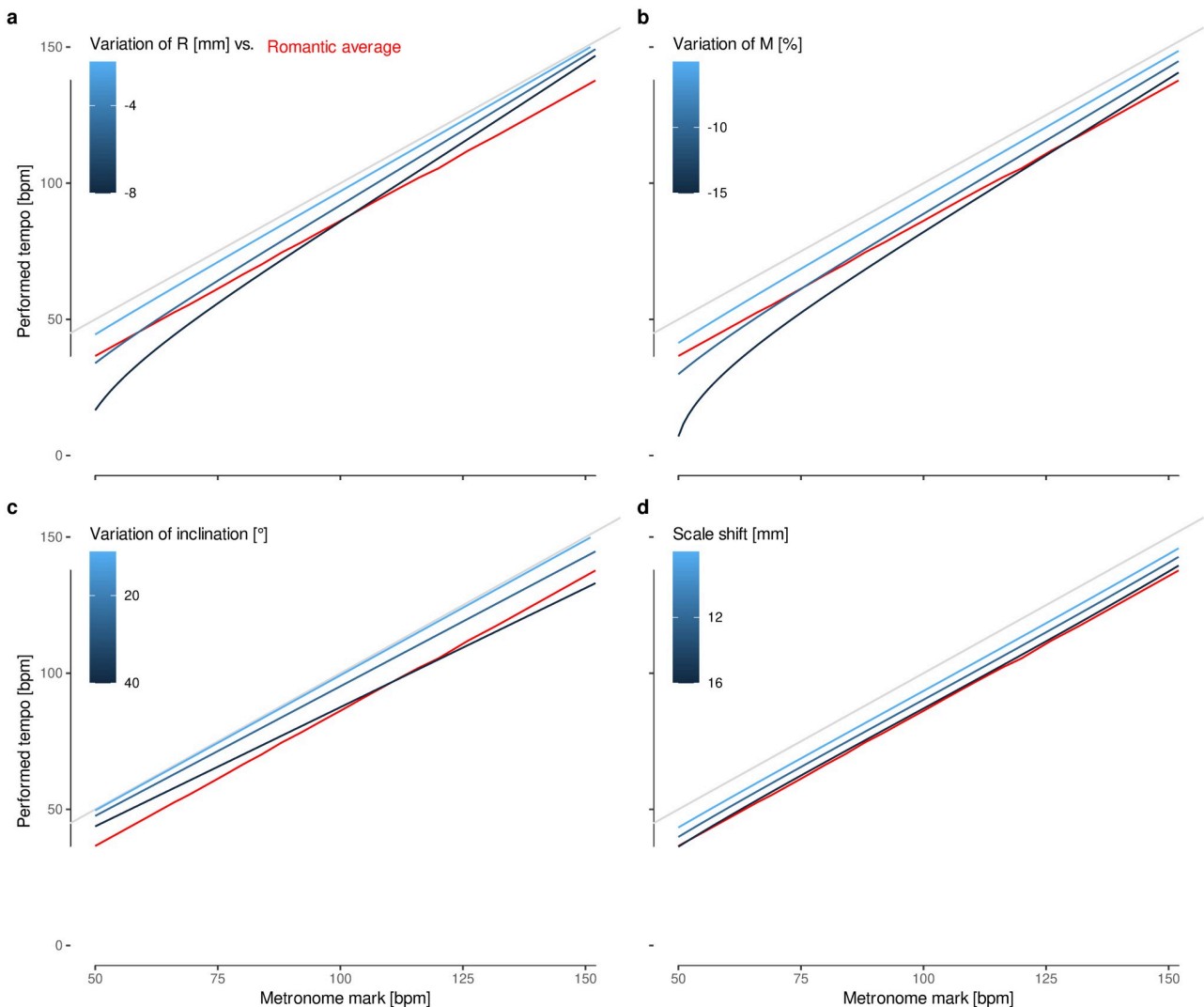

**Fig 10. Effect of different metronome distortions on its frequency compared to the average slow-down of Romantic conductors. a**, Reduction of the distance of the lower mass to the shaft, *R*. **b**, Reduction of the lower mass, *M*. **c**, Inclination of the metronome. **d**, Displacement of the scale relative to the moving weight.

## Acknowledgments

We thank Álvaro Perea-Covarrubias for his decisive support in the early stages of this research; and, especially, to Heidi and Peter Stadlen, who devoted several decades to putting together most of the pieces of this fascinating puzzle.

## Author Contributions

**Conceptualization:** Almudena Martin-Castro, Iñaki Ucar.

**Data curation:** Almudena Martin-Castro.

**Formal analysis:** Almudena Martin-Castro.

**Investigation:** Almudena Martin-Castro, Iñaki Ucar.

**Methodology:** Almudena Martin-Castro, Iñaki Ucar.

**Software:** Iñaki Ucar.

**Supervision:** Iñaki Ucar.

**Validation:** Almudena Martin-Castro, Iñaki Ucar.

**Visualization:** Almudena Martin-Castro, Iñaki Ucar.

**Writing – original draft:** Almudena Martin-Castro, Iñaki Ucar.

**Writing – review & editing:** Almudena Martin-Castro, Iñaki Ucar.

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
