## [Decision Letter · Decision Letter 0]

20 Nov 2020

PONE-D-20-33396

Conductors’ tempo choices shed light over Beethoven’s metronome

PLOS ONE

Dear Dr. Ucar,

Thank you for submitting your manuscript to PLOS ONE. After careful consideration, we feel that it has great merit but does not fully meet PLOS ONE’s publication criteria as it currently stands. Therefore, we invite you to submit a revised version of the manuscript that addresses the minor points raised during the review process.

Having personally read the paper, its methodology and its conclusions in detail, I believe that I fully adhere to the evaluations of Reviewer #1, so I suggest the authors to make the minor suggested changes, in this final round of review.

We look forward to receiving your revised manuscript.

Kind regards,

Alice Mado Proverbio

Academic Editor

PLOS ONE

Journal Requirements:

Reviewers' comments:

Reviewer's Responses to Questions

**Comments to the Author**

1. Is the manuscript technically sound, and do the data support the conclusions?

Reviewer #1: Yes

2. Has the statistical analysis been performed appropriately and rigorously? 

Reviewer #1: Yes

3. Have the authors made all data underlying the findings in their manuscript fully available?

Reviewer #1: Yes

4. Is the manuscript presented in an intelligible fashion and written in standard English?

Reviewer #1: Yes

5. Review Comments to the Author

Reviewer #1: Review of paper PONE-D-20-33396

Conductors tempo choices shed light’s over Beethoven’s metronome

This paper analyzes the role of Maelzel metronomous ambigue scale in inducing the music composer Ludwig van Beethoven in choosing BPM tempos systematically higher than what he actually intended.

In order to do the authors analyzed the complete recordings of Beethoven’s symphonies performed by 36 different conductors from different styles and time periods, ranging from the 1940s to the 2010s and analyzed them automatically using a tempo estimation algorithm. They also recreated Beethoven’s very metronome from photographs.

Data analysis showed that all the conductors, including the one more adhering to metronomic indications (such as for example Riccardo Chailly), systematically deviated from Beethoven’s marks and used slower, or much slower tempos.

Overall, this is a very interesting paper, nicely written and accurately documented. It uses a rigorous and sound methodology, and the discussion of the results is fully supported by evidences and data analysis. We think that it makes an excellent contribution to both musicology and history of science and measurements.

I have only a few suggestion outlined below:

1) Top of page 3 (line 70), Results. Please spend a few more words about the methods for classifying conductors in the 3 classes

2) Legend of Figure 3, please define CI (with a 95% CI)

3) Fig. 4b. Please indicate where this document is stored (Library, museum etc..)

4) Please define LRT, line 267, page 9 a significative random effect (LRT =

5) Fig 2 and Fig. 4. Can you possible increase the contrast, rendering the gray more dark?

6. PLOS authors have the option to publish the peer review history of their article (what does this mean?). If published, this will include your full peer review and any attached files.

Reviewer #1: No

---

## [Author Response · Author response to Decision Letter 0]

23 Nov 2020

We are very grateful for a quick yet thorough review, as well as for the kind words about our work. In the following, we provide a response to each comment, explaining how it has been addressed in the new version of the paper. For convenience, we provide a revised version with tracked changes along with the untracked version.

1. Top of page 3 (line 70), Results. Please spend a few more words about the methods for classifying conductors in the 3 classes

Thanks for bringing this into our attention. The classification of performances is made by convention: HI are those that use period instruments and follow all the usual HI stylistic criteria (as analysed by Young [7] among others), and usually simply because the conductor explicitly stated the intention to follow such criteria; HI-influenced performers follow the same stylistic guidelines, but may be not so strict in terms of instrumentation; finally, Romantic performances are those previous to the 1980s or, more generally, those that do not adhere to HI performing criteria. These details can be found in the performance reviews included with the recordings (identified in Table 1) as well as Young’s PhD thesis [7].

Our intention was to provide a quick note in Results (line 70), and then this description in Methods (line 177), but we forgot to add it. Therefore, we have included this longer explanation to Methods (lines 177-184) in the revised version of the manuscript, because we believe it belongs in this section. However, if bringing such a paragraph forward to Results is considered best for the readership, we would be happy to move it. Note also that we have corrected a broken reference to Table 1 (line 178).

2. Legend of Figure 3, please define CI (with a 95% CI)

Added “Confidence Interval” to figure legend as well as line 274.

3. Fig. 4b. Please indicate where this document is stored (Library, museum etc..)

Thanks, we forgot to add the reference. The autograph of the 9th Symphony is stored in the Staatsbibliothek in Berlin, and it can be accessed online through a permanent URL. We have added this as reference number 28 in the revised version of the manuscript. We have also added 3 missing URLs in references 25-27.

4. Please define LRT, line 267, page 9 a significative random effect (LRT =

Added “Likelihood Ratio Test” to line 275 of the revised version of the manuscript.

5. Fig 2 and Fig. 4. Can you possibly increase the contrast, rendering the gray more dark?

We have increased the contrast in both figures, and in Fig 4b we have specified that this is an enhanced version of the image.

---

## [Editor Report · Decision Letter 1]

25 Nov 2020

Conductors’ tempo choices shed light over Beethoven’s metronome

PONE-D-20-33396R1

Dear Dr. Ucar,

We’re pleased to inform you that your manuscript has been judged scientifically suitable for publication and will be formally accepted for publication once it meets all outstanding technical requirements.

Kind regards,

Alice Mado Proverbio

Academic Editor

PLOS ONE
---

## [Editor Report · Acceptance letter]

27 Nov 2020

PONE-D-20-33396R1 

Conductors' *tempo* choices shed light over Beethoven's metronome 

Dear Dr. Ucar:

I'm pleased to inform you that your manuscript has been deemed suitable for publication in PLOS ONE. Congratulations! Your manuscript is now with our production department. 

Kind regards, 

on behalf of

Dr. Alice Mado Proverbio 

Academic Editor

PLOS ONE